# DEEP POSITIVE UNLABELED LEARNING WITH A SEQUENTIAL BIAS

## ABSTRACT

For many domains, from video stream analytics to human activity recognition, only weakly-labeled datasets are available. Worse yet, the given labels are often assigned *sequentially*, resulting in *sequential bias*. Current Positive Unlabeled (PU) classifiers, a state-of-the-art family of robust semi-supervised methods, are ineffective under sequential bias. In this work, we propose *DeepSPU*, the first method to address this sequential bias problem. *DeepSPU* tackles the two interdependent subproblems of learning both the latent labeling process and the true class likelihoods within one architecture. We achieve this by developing a novel iterative learning strategy aided by theoretically-justified cost terms to avoid collapsing into a naive classifier. Our experimental studies demonstrate that *DeepSPU* outperforms state-of-the-art methods by over 10% on diverse real-world datasets.

## 1 INTRODUCTION

**Motivation.** State-of-the-art approaches for learning from data with only only incomplete positive labels require an accurate estimation of the likelihood that any given positive instance receives a label, known as the *propensity score*. However, all existing approaches overlook the fact that the annotations given for sequential data are often clustered together, and thus the likelihood that a given instance is labeled is dependent on the labels of the surrounding instances. We refer to this as *sequential bias*. Overlooking this sequential bias results in an incorrect propensity score and significantly reduced classification performance. Ours is the first work to make this observation and we propose the first solution to this open problem.

*Human Activity Recognition* (HAR) is a prime example of sequential bias in data. To collect HAR data, subjects are asked to report their activities while wearing mobile sensors. As study-length increases (collection may take many days), participants leave many activities unlabeled. Additionally, wearable sensors record data rapidly so large blocks of time get labeled consecutively, also creating sequential bias. Many more applications, such as intrusion detection from video or illness prediction from medical records, have similar sequentially-labeled data are are susceptible to sequential bias (Rodríguez-Moreno et al., 2019; Schaekermann et al., 2018). This is a crucial issue as existing methods show drastically reduced accuracy when sequential bias is not accounted for (as demonstrated in our Experimental Results).

**State-of-the-Art.** Positive Unlabeled (PU) classifiers are a family of semi-supervised methods that learn from incompletely-labeled data without requiring *any* labeled negative examples (Bekker & Davis, 2020; Elkan & Noto, 2008; Li & Liu, 2005; Hsieh et al., 2015; Du Plessis et al., 2015; Kiryo et al., 2017; Bekker & Davis, 2018a; Kato et al., 2019). This is a key strength of PU methods because representative negative examples, typically required by semi-supervised methods, are often not feasible to acquire. For instance, in the HAR example, there are infinitely many activities that an individual is *not* performing at any given time. Consequentially, participants are only expected to provide some *positive* labels for their activities (Vaizman et al., 2017).

Unfortunately, existing PU methods make unrealistically restrictive simplifying assumptions on how the labels were applied. Specifically, they either assume that there is no bias in the labeling process (the probability of a sample being an unlabeled positive instance is uniform) (Elkan & Noto, 2008; Du Plessis et al., 2015; Kiryo et al., 2017) or otherwise only depends on the local attributes of each instance (Bekker & Davis, 2018a; Kato et al., 2019). This means that existing methods do not

model *sequential bias*. And, as we demonstrate in our experiments, these methods are significantly negatively impacted when a sequential bias is present.

**Problem Description and Technical Challenges.** Given a dataset of sequences, our goal is to predict the true class likelihoods of each instance within the given sequence given a subset of labeled positive instances during training. In particular, we focus on the difficult case where labels have been assigned with a *sequential bias*, with sequential bias defined as the case where the likelihood that a positive instance is labeled varies depending on whether its neighboring instances were labeled.

This problem is challenging due to two difficult interdependent subproblems. First, we have the *dependency* problem: if we had a model of the latent labeling process (which we call the *propensity model*) that allowed us to identify the true unlabeled positive instances, then we could use this propensity model to train a classifier to produce the true class likelihoods. However, we need these same true class likelihoods in order to train the propensity model - causing a cyclic dependency. Second, standard maximum likelihood estimation inherently assumes all instances are labeled, leading to a naive classifier in the presence of labeling bias. To capture unlabeled positive instances, a PU classifier must instead predict an *appropriate* number of positive instances without simply assuming all positives instances are labeled.

**Our Approach: DeepSPU.** We propose *Deep Sequential PU* (*DeepSPU*), which is the first Positive Unlabeled method to use a propensity score model that predicts the likelihood that any given positive instance is labeled while taking sequential bias into account. The propensity score allows us to train a classifier network given only partially labeled data. We achieve this by developing a novel learning method that overcomes the cyclic dependency problem by iteratively learning the propensity score model and the classifier using weakly-labeled data. Further, we introduce the two novel PU cost terms: the Prior-Matching Costs (PMC) and the Observation-Matching Costs (OMC), which prohibit the propensity model and classifier from collapsing into incorrect naive solutions.

**Contributions.** The main contributions of our work are:

- We identify *sequential bias*, a labeling pattern characteristic of many real-world labeling processes, and demonstrate how ignoring this bias significantly impacts the performance of state-of-the-art PU classifiers.

- We propose the first learning strategy to minimize the bias incurred from sequentially biased PU data. Namely, we propose an iterative learning strategy and design two novel PU cost terms, Prior-Matching and Observation-Matching, which prohibit collapse into certain incorrect adversarial solutions, as justified through theoretical analysis.

- We develop DeepSPU, the first model to mitigate sequential bias. DeepSPU uses the aforementioned learning strategy to jointly estimate the two interdependent latent variables: the propensity score and the true class probabilities - without any direct supervision for either learning task.

## 2 RELATED WORK

PU classifiers, a special type of semi-supervised models, learn from weakly-labeled training data without requiring any labeled negative examples (Bekker & Davis, 2020). In contrast, semi-supervised methods lean heavily on having access to both labeled positive and labeled negatives to learn from weakly supervised data, making them less robust than PU alternatives (Van Engelen & Hoos, 2020). PU methods are also more robust than a similar family of classifiers known as One Class (OC) classifiers, as general OC classifiers do not make use of unlabeled data during training (Khan & Madden, 2009).

There are many approaches to PU learning, such as re-weighting predictions (Zhang & Lee, 2005; Elkan & Noto, 2008), iteratively identifying reliable examples (Ienco & Pensa, 2016), and most notably risk minimization (Northcutt et al., 2017; Du Plessis et al., 2015; Kiryo et al., 2017). However, all these state-of-the-art approaches share the often-unrealistic assumption that *no bias* exists in the labeling process, sequential or otherwise. When a bias is present, these methods are susceptible to learning skewed decision boundaries and thus are prone to making biased and incorrect classifications (Bekker & Davis, 2020).

A few recent PU methods have begun to model bias in the labeling process, though none have captured *sequential* bias. One approach assumes the likelihood that a given true positive instance is labeled depends solely on its distance from the negative distribution (Kato et al., 2019), while others explore the case when this labeling likelihood depends on the general position of the positive instance in the feature space (Bekker & Davis, 2018a). Our DeepSPU method extends beyond these methods by coping with *both* this feature-level bias *and* the previously-overlooked sequential bias.

## 3 PROBLEM SETTING

### 3.1 POSITIVE UNLABELED LEARNING

Positive Unlabeled (PU) learning is the task of training a classifier to predict the *true class* of each instance given a set of mostly unlabeled data and only some *labeled positive* instances (Bekker & Davis, 2020). Solving this problem requires modeling the likelihood that any given true positive instance is labeled. This labeling likelihood is referred to as the *propensity score* (Bekker & Davis, 2018a). Estimating the propensity score is *not* the same as predicting whether an instance is labeled, which is often relatively easy. This is because estimating the propensity score corresponds to determining the likelihood that an instance is labeled *conditioned on the instance being a true positive*. We cannot directly estimate this value from the data as *we do not know whether or not any given unlabeled instance is a true positive*.

Formally, let $\mathcal{D} = \{(X^{(j)}, \mathcal{L}^{(j)})\}_{j=1}^{N}$ be a dataset $\mathcal{D}$ of $N$ sequence pairs, where $X^{(j)}$ is a sequence of real values and $\mathcal{L}^{(j)}$ is a sequence of binary label indicators, $|X^{(j)}| = |\mathcal{L}^{(j)}|$. For readability, we drop the superscript $j$ and describe our approach in terms of one sequence. Let $X = (x_1, x_2, ..., x_T)$ be a sequence of $T$ real values (which we refer to as a sequence of *instances*), and $\mathcal{L} = (\ell_1, \ell_2, ..., \ell_T)$ be an associated sequence of label indicators such that $\ell_i = 1$ if $x_i$ is labeled positive, and is 0 otherwise. Additionally, for each feature-label sequence pair there is an *unobserved* binary true class sequence, $Y = (y_1, y_2, ..., y_T), y_i \in \{0, 1\}$, representing the underlying classes of the instance. In addition to $y_i$ being unavailable during training, $Pr(y_i = 1|\ell_i = 1) = 1$ as we assume there are no labels for negative instances, while $Pr(\ell_i = 1|y_i = 1) \neq 0$ as we assume not all positive instances are labeled.

We consider both *feature-level* and *sequential* biases in the labeling process. Feature-level bias assumes the propensity score for a positive instance depends on local features of the instance and thus $Pr(\ell_i = 1|y_i = 1) \neq Pr(\ell_i = 1|y_i = 1, x_i)$. Sequential bias assumes the propensity score of a positive instance can also depend on the label status of preceding instances, $\ell_{1:i-1}$, and thus $Pr(\ell_i = 1|y_i = 1) \neq Pr(\ell_i = 1|y_i = 1, x_{1:i}, \ell_{1:i-1})$. We call a propensity score that captures sequential and feature-level bias a *sequential propensity score* $q_i = Pr(\ell_i = 1|y_i = 1, x_{1:i}, \ell_{1:i-1})$.

Our goal is to train a classifier $g_\theta(\cdot)$ with parameters $\theta$ to solve the binary classification problem, such that $g_\theta(X) = Pr_Y(Y|X)$. During training, only the features $X$ and label status indicator $\mathcal{L}$ are observed while the true class $Y$ is unobserved. Notation used summarized in Appendix A.6.

### 3.2 BACKGROUND ON EMPIRICAL PU RISK MINIMIZATION

In standard positive-negative binary classification, the *risk* of a classifier $g_\theta$ is given as:

$$R(g_\theta) = \pi \mathbb{E}_p[C^+(g(x))] + (1 - \pi)\mathbb{E}_n[C^-(g(x))],$$

where $\mathbb{E}_p$ and $\mathbb{E}_p$ are the distribution over the positive and negative instances respectively, and $C^+$ is the loss incurred from predicting $g_\theta(x)$ given that the true instance is positive while $C^-$ is the loss incurred from predicting $g_\theta(x)$ when the true instance is negative. Several recent works have focused on reformulating the above risk into a "positive-unlabeled" risk that takes expectations over the labelled and unlabeled distributions rather than the positive and negative distributions, as the latter two distributions can not be estimated directly from PU data (Kiryo et al., 2017; Du Plessis et al., 2015). Directly minimizing the empirical PU risk has been successful in the unbiased SCAR setting (Du Plessis et al., 2015; Kiryo et al., 2017)

Additionally, Bekker *et al.* proposed a PU risk for the SAR setting, where there is bias in the labeling that is a function of the feature values of each instance (Bekker & Davis, 2018a). For classifier $g_\theta$

this risk is given as:

$$R(g_\theta) = \pi c \mathbb{E}_\ell \left[ \frac{1}{e(x)} C^+(g(x)) + \left( 1 - \frac{1}{e(x)} \right) C^-(g(x)) \right]$$
$$+ (1 - \pi c) \mathbb{E}_u [C^-(g_\theta(x))], \tag{1}$$

where $e(x)$ is the *propensity score* and represents the probability that a positive instance is labeled. In the formulation proposed by Bekker *et al.* the propensity score is *only* a function of the local feature values $x$.

A classifier that directly minimizes Bekker's PU risk has not been proposed for the case where both the propensity core $e(x)$ and the posterior $P_Y(Y|X)$ are unknown. This is due to the difficulty of estimating both the propensity score and the posterior jointly, which arises from the fact that in the above risk the perceived performance of the estimated posterior $g_\theta$ (and thus the gradients incurred) is based on the estimate of the propensity score and vice versa. Ergo, a poor estimate of the propensity score can lead the classifier $g_\theta$ to a poor solution while a poor classifier $g_\theta$ leads to an inaccurate estimator of the propensity score.

As described in the following section our proposed *DeepSPU* method does succeed in learning the propensity score and $g_\theta$ jointly by minimizing the PU risk directly. *DeepSPU* overcomes the aforementioned difficulty of training the two latent variables through risk minimization by using a novel iterative training algorithm coupled with additional regularization terms.

## 4 METHODOLOGY

### 4.1 OVERVIEW

We now describe a general estimation procedure for sequentially biased PU data, along with a specific model for learning in this setting.

### 4.2 SEQUENTIAL BIAS LEARNING STRATEGY

There are 2 major components to our proposed learning strategy. First, we employ an innovative Iterative Learning Strategy, iteratively training a classifier model and a propensity score model by minimizing the *positive unlabeled risk* (Bekker & Davis, 2020). This is achieved without explicit feedback of the two estimated latent variables. Second, we design two novel PU cost (regularization) terms that are employed during the iterative training. These cost terms prevent the networks from converging on naive solutions which minimize the PU risk but do not result in correct probability distributions for the latent target variables.

#### 4.2.1 ITERATIVE LEARNING STRATEGY

As stated in section 3.2 typical classifiers for fully-labeled data are trained to minimize the expected value of a loss function $C$, known as the *risk*. If the propensity score is known then the standard risk $R$ can be expressed in terms of expectations over only positive and unlabeled distributions, instead of the positive and negative distributions (Du Plessis et al., 2015; Bekker & Davis, 2018a). Thus, to train our model we express the *empirical positive unlabeled risk*, $R_{\text{PU}}$, in terms of our novel sequential propensity score $q_i$ as:

$$R_{\text{PU}}(g,q|X,\mathcal{L}) = \frac{1}{T} \left( \sum_{x_i|\ell_i=1} \left( \frac{1}{q_i} C^+(g(x_i)) + \left( 1 - \frac{1}{q_i} \right) C^-(g(x_i)) \right) + \sum_{x_i|\ell_i=0} (C^-(g(x_i))) \right), \tag{2}$$

where $C^+$ is the loss incurred from predicting $g(x)$ assuming that the true class is positive, $C^-$ is the loss incurred from predicting $g(x)$ assuming the true class is negative, and $q_i = Pr(\ell_i = 1|x_{1:i}, \ell_{1:i-1})$ is the propensity score of the $i$-th instance. If our propensity scores are accurate, then minimizing the above equation will correspond to minimizing the true risk. This means that in effect we can learn the same classifier that we would have found if we had been given fully labeled data.

**Theorem 1.** *If $q_i = Pr(\ell = 1|y_i = 1, x_{1:i}, \ell_{1:i-1})$ for $i = 1 : T$, then $R_{pu}(g, q|X, \mathcal{L})$ is an unbiased estimation of the true risk $R(g|Y)$.*

A proof of this theorem is given in Section A.4 of the Appendix. As stated, Theorem 1 indicates that we can train a classifier on positive and unlabeled data if we have accurate propensity scores. However, the propensity score is in general an unobserved observed variable to use as a target during training, and thus there is no straightforward way to minimize the PU risk. The naive approach would be to simply learn the parameters of the classifier $g$ and propensity model $q$ simultaneously by minimizing Equation 2. However, there is no guarantee that Equation 2 is an unbiased estimate of the true risk if the estimated propensity scores are incorrect (and thus no indication that this would yield a good classification model). We propose to overcome this challenge with a novel *Iterative Learning Strategy*, outlined below.

We begin training the a classification model and propensity score model by initializing the propensity scores with "good" estimates, acquired by assuming the labeling likelihood $c = Pr(\ell_i = 1|y_i = 1)$ is constant for all positive instances. $c$ is easily computed given a prior on the class $\pi = Pr(y = 1)$ (Bekker & Davis, 2020), and has been shown to be successfully estimated from the initial data set (Jain et al., 2016). We show how to derive $c$ from $\pi$ in the appendix.

Next, we train a propensity model and a classification model iteratively in separate interleaved stages. That is, at each stage of training, either the parameters of the propensity model or classification model are updated independently from each other.

### 4.2.2   REGULARIZING COST TERMS

It is possible to minimize Equation 2 by naively setting $q_i = 1 \ \forall \ i$ and $g(x_i) = \ell_i$. In this case, the classifier does not predict the instance's *true class*, and instead erroneously predicts whether or not an instance is labeled (Bekker & Davis, 2018a). To avoid this adversarial solution, we propose two cost terms. First, we add the *Prior-Matching Costs (PMC)*, which drives the percentage of predicted positives to match the percentage of true positive we expect according to our prior. The PMC is given by the KL divergence between the unconditioned distribution of predicted positives and $Bern(\pi)$, the Bernoulli distribution parameterized by the true class prior $\pi$:

$$\text{PMC}(X) = \text{KL}(Pr(\hat{y} = 1|\Theta)||Bern(\pi)). \tag{3}$$

Thus, the number of predicted positives are driven to match the expected number of true positives. By definition, the PMC will be larger than the number of labeled instances, and thus Equation $T1$ will incur a high penalty cost for the *adversarial solution*.

Additionally, if the estimated propensity scores match the true propensity scores, then our propensity score multiplied by the class probability of the $i$-th instance will equal the probability that the $i$-th instance is labeled, as stated in Theorem 2. This informs our next cost term.

**Theorem 2.** *If $q_i = Pr(\ell_i = 1|x_{1:i}, \ell_{1:i-1}, y_i = 1)$ is the sequential propensity score of the $i$-th instance, then $Pr(\ell_i = 1|x_i, \ell_{i-1}) = q_i \cdot Pr(y_i|x_i)$.*

The proof is given in Appendix A.5.

In short, we propose to explicitly encourage the product of the estimated propensity scores and predicted class probabilities to match the observed labels. We accomplish this by adding the Binary Cross Entropy (BCE) between them as next cost term. We refer to this as the *Observation-Matching Costs (OMC)*, as it requires our propensity score and class predictions to match the observed data:

$$\text{OMC}(X) = -\sum_{j=1}^{N}\sum_{i=1}^{T} \ell_i^{(j)} log\left(\hat{y}_i^{(j)} q_i^{(j)}\right) \tag{4}$$

We thus propose a final combined cost function sequentially biased PU data as follows:

$$L(\Phi, \Theta) = I \cdot J_{Q_\phi}(g_\Theta) + (1 - I) \cdot J_{g_\Theta}(Q_\phi),$$

where $I$ is an indicator variable that equals 1 if we're updating the parameters of the classification model $g_\Theta$ and is 0 otherwise, and $J$ is the cost term defined as:

$$J_a(b) = R_{PU} + \lambda_1 \text{PMC} + \lambda_2 \text{OMC}, \tag{5}$$

with $\lambda_1$ and $\lambda_2$ being weights on the corresponding cost terms and $J_a(b)$ corresponding to the above equation as a function of $b$ while $a$ is held constant.

### 4.3 THE DEEPSPU MODEL FOR SEQUENTIALLY BIASED DATA

We now propose a specific model to minimize Equation 5. Our model, *DeepSPU*, consists of three sub-networks: 1) the *Representation Network*, which learns a robust representation of the input data, 2) the *Sequential Propensity Network*, which models the likelihood that an instance is labeled given that it is positive, and 3) the *Classification Network*, which models the likelihood that a given instance belongs to the positive class. The Sequential Propensity Network, used only during the training stage, is crucial in the training of the Classification Network as it allows us to train the Classification Network given only weakly-labeled training data. After training, the Classification Network can be deployed without the Sequential Propensity Network to predict the true class of new instances. Detailed pseudo-code is available in the appendix.

#### 4.3.1 REPRESENTATION NETWORK

The Sequential Propensity Network and the Classification Network both share a common base representation of the input data. This shared representation is modeled by a recurrent neural network (RNN) $B_\Omega$ with parameters $\Omega$, such that $B_\Omega$ takes in the sequence of input data $X = (x_1, x_2, \cdots, x_T)$ and maps each element of the sequence to a corresponding latent representation $H = (h_1, h_2, \cdots, h_T)$. Each latent representation $h_i$ is given by $h_i = F_h(x_i, h_{i-1})$, with the specific form of $F_h$ determined by the choice of RNN network. Our implementation of *Deep-SPU* uses a Gated Recurrent Unit (GRU) for the RNN (Cho et al., 2014), though the same principles apply to other RNN architectures.

### 4.4 SEQUENTIAL PROPENSITY NETWORK

The aptly-named Sequential Propensity Network models the propensity score conditioned on the local feature values of each instance and the feature values of all preceding instances, along with the label indicators of all preceding instances. The first component of this sub-network is another GRU that for each label $\ell_i$ produces a latent representation $s_i$ conditioned on all previous labels. In effect, $s_i$ summarizes the label subsequence $\ell_1$ to $\ell_i$ into one latent variable. This GRU is coupled with a feed-forward network (FFN) that takes in the latent label representation $s_{i-1}$, the previous label indicator $\ell_{i-1}$, and the representation of the input features learned by the Representation Network $h_i$ in order to produce the propensity score $q_i$ of the $ith$ instance. With $\Phi$ representing the learnable parameters of the Sequential Propensity Network $Q_\Phi$, $Q_\Phi$ models the sequential propensity score as $Q_\Phi(h_i, s_{i-1}, \ell_{i-1}) = q_i = Pr(\ell_i = 1 | x_{1:i}, \ell_{1:i-1})$.

### 4.5 CLASSIFICATION NETWORK

The shared hidden representation $h_i$ is also passed as input into the *DeepSPU classifier subnetwork*, which models the positive class probability given the features of the input data. Our classifier model $g_\Theta(x_i) = P(y_i | x_{1:i}, \Theta)$ with parameters $\Theta$ is given by: $g_\Theta(x) = F_g(B_\Omega(x_i); \Theta)$, where $F_g$ is a fully connected network with parameters $\Theta$. The predicted class value for an instance $x_i$ is given by $round(g(x_i))$, where $round(z) = 1$ if $z > 0.5$ and is 0 otherwise. We chose 0.5 as the cutoff as $g_\Theta(x_i)$ represents the *probability* that $y_i = 1$.

## 5 EXPERIMENTS

### 5.1 DATA AND EXPERIMENTAL DETAILS

**Datasets.** We evaluate our models and a relevant set of baselines on several real-world sequential human activity recognition datasets: UCI HAR[1] (Anguita et al., 2013), Older Healthy (OH) HAR[2] (Torres et al., 2013), and ExtraSensory (ES)[3] (Vaizman et al., 2017). ExtraSensory was subsampled at one reading for every 10 minutes of collected data due to the size of the dataset. Due to extreme class imbalance, we combine the multiple classes into one for some of the datasets.

---

[1] https://archive.ics.uci.edu/ml/datasets/human+activity+recognition+using+smartphones
[2] https://archive.ics.uci.edu/ml/datasets/Activity+recognition+with+healthy+older+people+using+a+batteryless+wearable+sensor
[3] http://extrasensory.ucsd.edu/

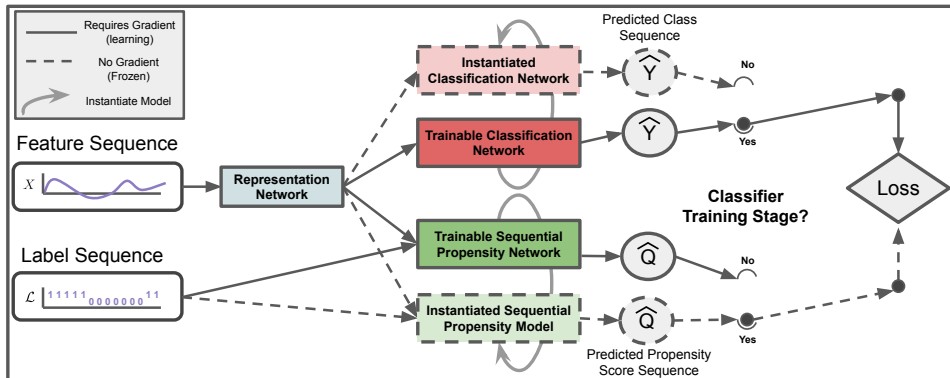

Figure 1: An overview of the DeepSPU learning process. The Representation Network learns a hidden representation of the input sequence that is fed to both the Classification Network and the Sequential Propensity Network. Networks are alternately trained iteratively, with one being frozen while the other is updated.

We combine `UCI HAR` into 3 classes: one representing Walking-related activities, one that combines Going Up and Down Stairs, and one that combines stationary activities. We likewise combine stationary activities in `OH HAR`. We also evaluate on `RealityCommons (RC Flu)`[4] (Madan et al., 2011), an in-the-wild health dataset, where the task is to classify whether the individual is experiencing flu symptoms on a given day using mobile sensor data.

**Compared methods.** We compare DeepSPU to the following state-of-the-art methods:

- *Positive-Negative (PN) Classifier* (Cho et al., 2014): As a baseline, we adopt a standard binary positive-negative classifier that treats all unlabeled instances as negatives. The model otherwise has the same structure as the DeepSPU classification network.

- *uPU* (Du Plessis et al., 2015): uPU is a recent highly-influential approach to training deep networks on PU data. uPU's convex approach minimizes the empirical PU risk. However, this approach assumes *the propensity score is constant*.

- *nnPU* (Kiryo et al., 2017): Similarly to *uPU*, *nnPU* also assumes that the propensity score is constant to minimize the empirical PU risk. Additionally, nnPU clips the risk of the unlabeled instances at 0 to avoid infinitely negative risks and is thus liable to overfit during training.

- *SAR-EM* (Bekker & Davis, 2018a): SAR-EM is the leading PU method for learning under *feature-level biased labeling*. SAR-EM jointly trains a classifier and propensity network using an Expectation-Maximization algorithm, such that the propensity network can handle feature-level but *not* sequential bias.

We do not compare against any general semi-supervised methods. *This is not an oversight*. As per our problem definition, we assume we have no labeled negative examples. This means general semi-supervised methods are inapplicable. (Van Engelen & Hoos, 2020).

**Evaluation metric.** To measure the performance of our compared methods, we use *balanced accuracy* (BA) (Brodersen et al., 2010), which is defined as $\frac{1}{2} * \left( \frac{TP}{TP+FN} + \frac{TN}{TN+FP} \right)$. BA is similar to the standard accuracy metric, but accounts for class imbalance: a BA of 0.5 is achieved by a naive classifier, regardless of class imbalance. An optimal classifier will achieve a BA of 1.0.

**Implementation details.** We use an 70%/10%/20% train/validation/test split for each dataset. The base classifier for all methods is a GRU with a 10-dimensional hidden layer. A 1-layer feed-forward neural network uses the GRU's latent representation as input for each instance and serves as the classifier. DeepSPU has an additional 1-layer GRU for the propensity network. Each method is trained until convergence (200 epochs). All methods are implemented in PyTorch and are publicly available [5].

---

[4] http://realitycommons.media.mit.edu/socialevolution4.html

[5] https://anonymous.4open.science/r/250569bb-b723-4e3c-8b37-ed982087c2db/

| Datasets: | ES Lying | ES Sitting | ES Walking | ES Sleeping | RealityCommons |
|-----------|----------|-----------|-----------|-------------|----------------|
| **PN:** | 0.68 | 0.60 | 0.62 | 0.66 | 0.50 |
| **uPU:** | 0.70 | 0.62 | 0.65 | 0.70 | 0.60 |
| **nnPU:** | 0.70 | 0.62 | 0.66 | 0.72 | 0.62 |
| **SAR EM:** | 0.66 | 0.63 | 0.58 | 0.69 | 0.54 |
| **DeepSPU:** | **0.77** | **0.66** | **0.68** | **0.78** | **0.66** |

Table 1: Performance of DeepSPU vs compared methods on naturally weakly labeled real-world datasets experiencing sequential bias. DeepSPU outperforms the others. Results reported as Balanced Accuracy. Higher is better.

| Method: | PN | | | uPU | | | nnPU | | | SAR-EM | | | DeepSPU | | |
|---------|----|----|----|-----|----|----|------|----|----|--------|----|----|---------|----|----|
| **% Labeled:** | 5 | 10 | 15 | 5 | 10 | 15 | 5 | 10 | 15 | 5 | 10 | 15 | 5 | 10 | 15 |
| **OH Stationary** | 0.50 | 0.50 | 0.51 | 0.50 | 0.50 | 0.50 | 0.50 | 0.50 | 0.50 | 0.50 | 0.50 | 0.50 | 053 | **0.53** | **0.55** |
| **OH Ambulating** | 0.50 | 0.50 | 0.50 | 0.50 | 0.50 | 0.50 | 0.50 | 0.51 | 0.52 | 0.50 | 0.50 | 0.50 | 0.50 | **0.52** | **0.54** |
| **UCI Walking** | 0.50 | 0.50 | 0.50 | 0.50 | 0.51 | 0.53 | 0.50 | 0.51 | 0.52 | 0.51 | 0.52 | 0.54 | **0.58** | **0.62** | **0.72** |
| **UCI Stairs** | 0.50 | 0.50 | 0.50 | 0.59 | 0.62 | 0.72 | 0.64 | 0.65 | 0.67 | 0.52 | 0.53 | 0.58 | **0.79** | **0.80** | **0.86** |
| **UCI Stationary** | 0.50 | 0.51 | 0.53 | 0.91 | 0.91 | 0.93 | 0.91 | 0.91 | 0.94 | 0.79 | 0.79 | 0.85 | **0.96** | **0.96** | **0.97** |
| **ES Walking** | 0.50 | 0.50 | 0.50 | 0.50 | 0.50 | 0.50 | 0.50 | 0.50 | 0.50 | 0.50 | 0.50 | 0.50 | **0.51** | **0.52** | **0.54** |
| **ES Sitting** | 0.50 | 0.50 | 0.50 | 0.50 | 0.50 | 0.52 | 0.50 | 0.50 | 0.52 | 0.53 | 0.53 | 0.54 | **0.55** | **0.58** | **0.60** |
| **ES Sleeping** | 0.50 | 0.50 | 0.50 | 0.50 | 0.50 | 0.50 | 0.50 | 0.50 | 0.51 | 0.50 | 0.50 | 0.50 | **0.58** | **0.65** | **0.66** |
| **ES Lying** | 0.50 | 0.50 | 0.51 | 0.50 | 0.50 | 0.51 | 0.50 | 0.51 | 0.53 | 0.50 | 0.50 | 0.50 | **0.62** | **0.69** | **0.72** |

Table 2: Classification performance for various levels of labeling on the Older Healthy, UCI HAR, and ExtraSensory HAR datasets. Results reported as Balanced Accuracy (higher is better).

## 5.2 EXPERIMENTAL STUDY ON CLASSIFYING NATURALLY-UNLABELED DATA

First, we demonstrate *DeepSPU*'s ability to classify data that naturally exhibits sequential bias. This can be clearly shown for the `ExtraSensory` and `RealityCommons` datasets, because both were collected "in the wild" with study participants labeling their own data sequentially. Thus they are thus naturally weakly labeled. Further, due to the study design it is unclear if an instance that is not labeled positive is in fact negative or an unlabeled positive (Chang et al., 2017; Madan et al., 2011), thus fitting the positive unlabeled problem description. For this experiment, we estimate the class prior for *nnPU* and *DeepSPU* using *TIcE* (Bekker & Davis, 2018b). As shown in Table 1, *Deep-SPU* significantly outperforms the state-of-the-art methods for both datasets. This demonstrates that tackling sequential bias inherent in these real-world datasets is impactful. Additionally, the baseline underperforms the PU methods in almost all cases, highlighting the importance of leveraging PU classifiers for these weakly-labeled real-world problems. Overall, these results confirm that the pervasive problem of sequential bias in labeling can be mitigated by *DeepSPU*.

## 5.3 EXPERIMENTAL STUDY ON LEARNING FROM VARIOUS LEVELS OF LIMITED LABELS

In practice, the percentage of labeled positives available during training will vary from dataset to dataset. Thus, we perform this next experiment to study the impact of the proportion of labeling on each method's performance. In line with much of the recent PU work (Bekker & Davis, 2020; 2018a; Kiryo et al., 2017), we achieve this by removing subsets of labels from each dataset prior to training. We range label availability from 5% to 15% of all positive instances, and remove the labels from all negative instances. To encourage *sequential* bias, unlabeling is done sequentially: A binary Markov Chain decides whether or not to remove the label for each instance in turn. The likelihood the Markov Chain switches from "labeling" to "unlabeling" states is varied according to the desired level of unlabeling. Details of this unlabeling process are in Appendix A.7. As shown in Table 2, *DeepSPU* significantly outperforms all other methods for every level of unlabeling. This demonstrates that even with very few labels, modeling sequential bias leads to significant improvements in performance. As expected, the *Positive Negative* performs the worst, as it does not account for unlabeled positives. Surprisingly, *SAR-EM* is outperformed by *nnPU*, despite *nnPU* assuming that no bias arises in the labeling process. This may be explained by *nnPU*'s natural aversion to overfitting (Kiryo et al., 2017), while *SAR-EM* may learn spurious relationships in the labels while mistakenly modeling sequential bias as feature-level bias.

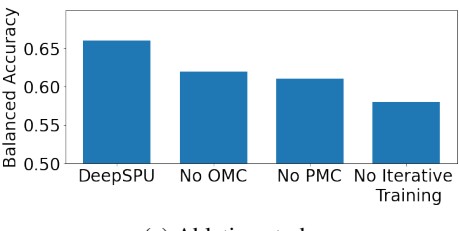
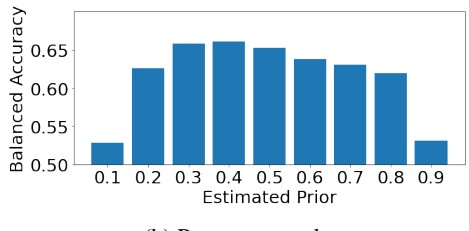

(a) Ablation study                    (b) Parameter study

Figure 2: a) The ablation study of *DeepSPU* indicates that each of its major components significantly improves its performance. b) Performance of *DeepSPU* varying values for the class prior $\pi$ estimate, with the true prior being about 0.36.

### 5.4 ABLATION STUDY

*DeepSPU* minimizes PU risk using three components: The iterative training algorithm, the *observation-matching cost* (OMC), and the *prior-matching cost* (PMC). In this experiment, we demonstrate the necessity of each component in an ablation study on the `ExtraSensory Sitting` dataset. Removing any of the three components results in significantly lower classification accuracy. Specifically, removing the iterative training strategy impacts the performance most significantly. This is expected, as without the iterative training strategy the cost function is likely to be biased. We also see that the PMC divergence, which encourages the percentage of predicted positives to match the class prior, is the more important of the two novel cost terms.

### 5.5 EXPERIMENTAL STUDY EVALUATING PARAMETERS FOR ESTIMATED PRIOR

PU methods rely on prior estimates of class label likelihoods $\pi = Pr(y = 1)$. In particular, *DeepSPU* uses this prior for the PMC term. With this value being estimated (Jain et al., 2016), the selected prior could be inaccurate in a practical setting. Therefore, we now study the impact of poorly-estimated priors on *DeepSPU* by varying this estimation. We train *DeepSPU* on the `ES Sitting` dataset, where the true class prior is about 0.36. As shown in Figure 2b, *DeepSPU* achieves the strongest performance when the estimated prior is closest to the true prior. However, even when the prior is overestimated, the performance is not significantly impacted until the estimated prior becomes severely over or underestimated. This indicates that in practice *DeepSPU* is generally robust even to incorrectly estimated priors.

## 6 CONCLUSION

We propose *DeepSPU*, the first PU solution learning from weakly labeled data with sequential bias in the labeling. We formulate a novel iterative learning strategy to jointly train a classification model and labeling likelihood (propensity) model, along with designing two theoretically-justified PU cost terms to account for this bias. Through a series of extensive experimental results we demonstrate that the previously-overlooked sequential labeling bias naturally arises in real-world datasets. Also, the state-of-the-art PU methods have poor performance when this type of bias is present, while *DeepSPU* achieves robust classification performance under sequential labeling bias for a rich variety of real-world data sets.

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

# A APPENDIX

## A.1 STRATEGIES FOR CALCULATING THE CLASS PRIOR

Our DeepSPU method and the state-of-the-art nnPU method both take in the class prior $\pi = p(y = 1)$ as a parameter. While this can be approximated from PU data (Jain et al., 2016), we calculated this from the true class labels for all experiments. This was done as follows:

$$\hat{\pi} = \frac{\sum_{n=1}^{N} \sum_{t=1}^{T} y_i^{(j)}}{NT},$$

where $N$ is the number of time series in the training set and T is the length of each series.

## A.2 DEEPSPU TRAINING

---

**Algorithm 1:** DeepSPU Training Process

---

**input** : Initialized propensity network $Q$
        Initialized classification network $G$
        Initialized shared representation $B$
        Dataset of sequences $\mathcal{D}$ with features $X$ and labels $\mathcal{L}$ of split into batches
        Prior estimate $\hat{\pi}$
**output:** Trained classifier
        Trained propensity function
$c = \frac{num\_labeled\_instances}{\hat{\pi}}$
 define $frozen\_propensity(x) = c$
 $train\_classifier = True$
 **for** $i \leftarrow 1$ **to** $max\_epochs$ **do**
    **for** $batch$ $in$ $num\_batches$ **do**
        **if** $train\_classifier$ **then**
            $\hat{Y}_{batch} = G(B(X_{batch}))$
            $\hat{Q}_{batch} = frozen\_propensity(frozen\_B(X_{batch}))$
        **else**
            $\hat{Y}_{batch} = frozen\_classifier(frozen\_B(X_{batch}))$
            $\hat{Q}_{batch} = Q(B(X_{batch}))$
        **end**
        $loss(\hat{Q}_{batch}, \hat{Y}_{batch}) = \hat{R}_{pu} + BCE + KL_{div}$
    **end**
    Update weights of $B$ with Adam optimizer using $loss$
    **if** $train\_classifier$ **then**
        Update weights of $G$ with Adam optimizer using $loss$
    **else**
        Update weights of $Q$ with Adam optimizer using $loss$
    **end**
    $train\_classifier = not\ train\_classifier$
    $frozen\_propensity = copy(Q)$
    $frozen\_classifier = copy(G)$
    $frozen\_B = copy(B)$
**end**

---

For the sake of reproducibility we provide the pseudocode for training DeepSPU.

### A.3 Finding Labeling Likelihood $c$ Given the Class Prior

As described in the Methodology section, DeepSPU initially considers the propensity score to be constant for the first few epochs of training. This constant propensity score is found given knowledge of the class prior $\pi$ as follows:

$$
\begin{aligned}
c &= Pr(\ell = 1 | y = 1) \\
&= \frac{Pr(\ell = 1, y = 1)}{Pr(y = 1)} \\
&= \frac{Pr(\ell = 1)}{Pr(y = 1)} \\
&= \frac{Pr(\ell = 1)}{\pi}
\end{aligned}
$$

Thus, we need only to estimate $Pr(\ell = 1)$ by finding the fraction of labeled instances in the dataset, and dividing this value by the class prior $\pi$.

### A.4 Proof of Theorem 1

*Proof.*

$$
\begin{aligned}
\mathbb{E}[R_{PU}(g, q | X, \mathcal{L})] &= \frac{1}{m} \sum_{i=1}^{m} y_i q_i \Big( \frac{1}{q_i} L^+(g(x_i)) + (1 - \frac{1}{q_i}) L^-(g(x_i)) \Big) + (1 - y_i q_i) L^-(g(x_i)) \\
&= \frac{1}{m} \sum_{i=1}^{m} y_i L^+(g(x_i)) + (1 - y_i) L^-(g(x_i)) \\
&= R(g|Y)
\end{aligned}
$$

$\square$

### A.5 Proof of Theorem 2

*Proof.*

$$
\begin{aligned}
Pr(\ell_i = 1 | x_{1:i}, \ell_{1:i-1}) &= Pr(\ell_i = 1 | x_i, \ell_{1:i-1}, y_i = 1) Pr(y_i = 1 | x_i, \ell_{1:i-1}) \\
&\quad + Pr(\ell_i = 1 | x_i, \ell_{i-1}, y_i = 0) Pr(y_i = 0 | x_i, \ell_{1:i-1}) \\
&= Pr(\ell_i = 1 | x_i, \ell_{i-1}, y_i = 1) Pr(y_i = 1 | x_i, \ell_{1:i-1}) \\
&= Pr(\ell_i = 1 | x_i, \ell_{i-1}, y_i = 1) Pr(y_i = 1 | x_i)
\end{aligned}
$$

$\square$

A.6 TABLE OF NOTATION

| Notation | Description |
|---|---|
| $N$ | Number of sequences in the dataset |
| $T$ | Length of sequence |
| $x_i$ | Features of $ith$ instance |
| $y_i$ | Binary class indicator variable for $ith$ instance $y_i = 1$ if $y_i$ is of the positive class, $y_i = 0$ otherwise. |
| $\ell_i$ | Binary label indicator variable for $ith$ instance. $\ell_i = 1$ if the timestep was labeled, $\ell_i = 0$ otherwise. |
| $q_i$ | Sequential propensity score of the $ith$ instance. $q_i = Pr(\ell_i = 1\|x_{1:i}; \ell_{1:i-1}, y_i = 1)$ |
| $\pi$ | Class prior. $\pi = Pr(y = 1)$ |
| $c$ | Label frequency. $c = Pr(\ell = 1\|y = 1)$ |
| Capital letter | Sequence of corresponding variable. i.e., $X = (x_1, \cdots, x_T)$ |
| where $i = 1, \cdots, N$ and $t = 1, \cdots, T$ | |

Table 3: Reference for symbols used in this work

| True Class (y) | 0 0 1 1 1 1 1 1 1 1 1 0 0 0 0 0 0 0 0 |
| Markov Chain | 1 1 0 0 0 0 0 0 |
| Labels (ℓ) | 0 0 1 1 0 0 0 0 0 0 0 0 0 0 0 0 0 0 0 |

Figure 3: An example of unlabeling with Markov chain.

## A.7 INTRODUCING SEQUENTIAL BIAS

For each sub-sequence of consecutive positive instances within each sequence, a binary Markov chain was run to decide which instances were to be unlabeled. We initiated the Markov chain at state '1', where state '1' indicates that the corresponding positive instance remains labeled. The Markov chain could transition to state '0' with probability $\lambda$. We varied the value of $\lambda$ to evaluate DeepSPU's performance for various levels of mislabeling. For instances where the Markov chain is in state '0' the corresponding positive instance was set to be unlabeled. The chain has a 1% probability of transitioning back to '1' from state 0. An example of this is shown in figure 3. This process introduces *sequential* bias, as the likelihood that a given instance is mislabeled depends on whether the previous instance is mislabeled (a given instance has a 100 X (1 - $\lambda$) % chance of being labeled if the preceding instance is labeled).

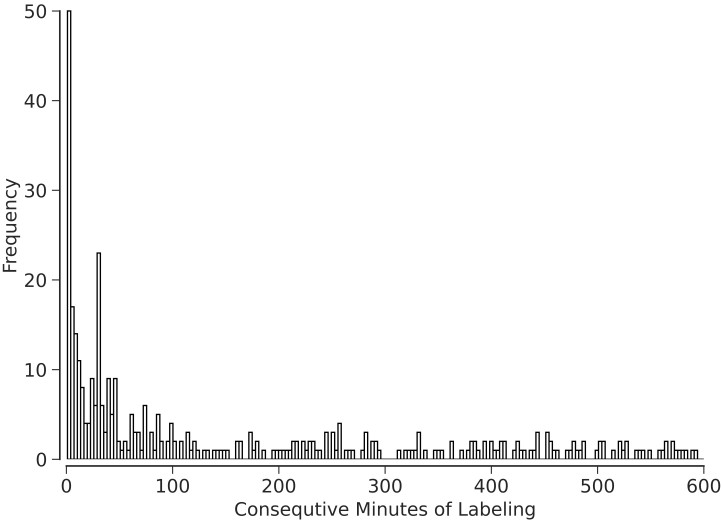

Figure 4: Histogram of lengths of consecutive labels (in minutes).

## A.8 QUANTITATIVE ANALYSIS ON EXTRASENSORY LABELING PATTERNS

Figure 4 shows the frequency of lengths of labeled subsequences in the ExtraSensory dataset. Clearly, the participants do enter phases of labeling as there are many long subsequences of consecutive labels.

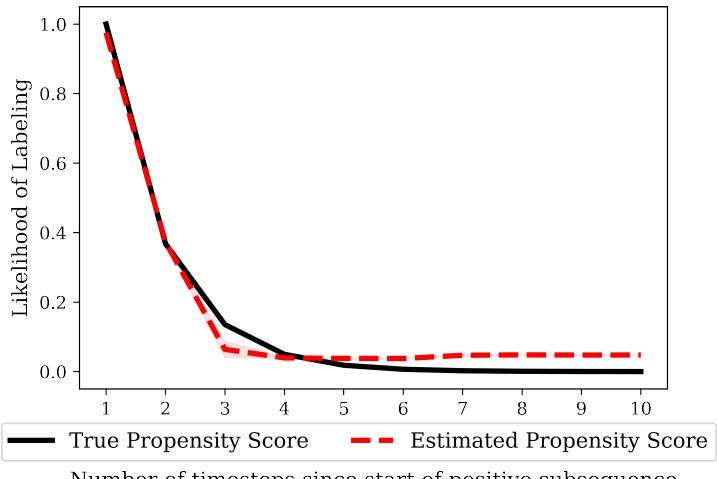

Figure 5: DeepSPU's estimated propensity score vs the true propensity scores. The estimated scores match the true scores almost perfectly.

### A.9    EVALUATING PERFORMANCE OF PROPENSITY MODEL

We perform an additional experiment in order to demonstrate DeepSPU's propensity network's ability to learn the true propensity score. In this experiment, we created subsequences of positive instances and subsequences of negative instances. The feature values for the positive instances where drawn from a normal distribution with mean 0 and unit variance, while the features for the negative instanced were drawn from a normal distribution with mean 10 and unit variance. The positive subsequences were labeled using a gamma distribution to decide which positive instances received labels. We used Scipy's (Virtanen et al., 2020) gamma distribution with shape, location, and scale parameters all set to 1. We then train DeepSPU on this data and extracted its learned propensity scores. As Figure 5 shows, the estimated propensity scores match the true propensity scores incurred by the gamma distribution almost perfectly.

