# OpenReview forum: "Deep Positive Unlabeled Learning with a Sequential Bias"
_ICLR.cc/2021/Conference — Reject_

### Official Review · AnonReviewer1 · 2020-10-30
**Ok paper, but writing and organization could be improved**

**Rating:** 6
**Confidence:** 4

**Review:**

--- Update after discussion ---

I thank the authors for considering my recommendations. I think the clarity of the updated paper is much improved, particularly the introduction. I further think that the authors have adequately addressed the concerns raised by the other reviewers and recommend accepting the paper.

--- Review ---

Overall, I thought this paper was good, but not great. I found the problem poorly motivated and the contributions limited in scope. The methods were reasonable and appear technically correct, if somewhat ad hoc. I recommend acceptance, though I think the paper could be substantially improved.

--- Comments ---

1. After reading the intro, it was very unclear to me what problem the paper was trying to solve and what the contributions of the paper were. In part, this is because the main problem addressed by the paper, "sequential bias", is not defined until the second page despite being reference earlier and in the abstract. Even the fact that we are dealing with sequential data wasn't made very clear until later. I recommend rewriting the intro to start by defining the problem that the paper solves, something like: Existing methods for learning instance-level classification models from sequential data with positive only labels rely on accurately estimating the conditional probability that an instance receives a label, known as the propensity score. However, these methods generally ignore the fact that positive labels are often temporally clustered together --- referred to as sequential bias --- resulting in inaccurate propensity scores and lower classification performance. We propose a method that ...

2. In section 3, I recommend giving more of a tutorial on how propensity scores are used in non-sequential settings. As it is, I wasn't sure how the propensity scores could be used to recover the true risk (and thus, why they were important) until page 5.

3. The main contribution of this paper doesn't really have anything to do with deep learning and I think tying the method so closely to NNs artificially limits the contributions. Specifically, most of section 4.5 and 4.6 could be presented without any reference to NNs. I recommend moving 4.5 and 4.6 before 4.1 - 4.3 and writing them as a general estimation procedure. 4.1 - 4.3 can then be framed as discussing the specific model specification details used in the paper. Even the name DeepSPU references deep learning when it doesn't really need to.

--- Minor comments ---

1. Page 2, par 2, line 2: "Instance" not yet defined.

2. Page 2, par 4, line 7: Prior-Matching --> Prior-Weighted

3. Page 3, par 3, line 2: I recommend $|\cdot|$ or $length(\cdot)$ rather than $||\cdot||$ to refer to the length of a sequence so as to avoid confusion with a norm.

4. Page 3, par 4, line 5: $y_i^{(j)}$ --> $y_i$

5. Page 3, par 4, line 8: I don't follow the logic that a low propensity score implies a high probability of being a true positive. Since we are conditioning on $x_i$, couldn't it also just be an instance that is obviously negative and thus unlikely to be labeled?

6. Page 3, par 5, line 3: Is there notation defined in the appendix that is not defined somewhere in the main text? If not, I'm not sure I would include this table.

7. Page 4, par 3, line 2: local feature values --> local feature values and the feature values of all preceding instances

8. The notation in the appendix differs from that used in the main paper and has several typos.

---

> ### Author Response · Authors · 2020-11-24
> **Thank you for your insightful feedback**
>
> Thank you for taking the time to provide these detailed suggestions on how to improve the presentation of work. We have taken your advice and restructured the paper as you recommended. Namely, we have rephrased the intro to make the focus of the work and the contributions clear in the first paragraph, modified the description of our contributions to convey that our proposed method is indeed viable even beyond just deep networks, added in more detail on PU risk minimization and propensity scores into the problem setting section (section 3), and restructured the methodology section to separate out our more general proposed optimization strategy from the details of our specific model. We thank you for these suggestions as they further improved our manuscript.

---

### Official Review · AnonReviewer6 · 2020-11-06
**TLDR; Interesting observation about sequential bias; not sufficiently substantiated (yet)**

**Rating:** 5
**Confidence:** 3

**Review:**

**Summary**: This paper makes the observation that sequential labelling processes introduce a bias in the obtained labels. This is relevant for weakly supervised sequential settings (eg human activity recognition), where examples are either assigned a positive label or are unlabelled. For this setting, a novel method is introduced that models this (latent) labelling process and uses the latent labels in the objective function,  this results in better performance on two datasets. While the observation of sequential bias is interesting and the performance is promising, I rate the current paper as a 'OK but not good enough': it seems to miss the insights behind the observation of sequential labelling bias and it seems to miss some experiments (ablation study, multi-class, etc).

**Sequential Labelling Bias**:
The observation about sequential bias is interesting and probably valid.
But some claims are doubtful without further backup by experiments or references, for example: "Hence, an instance is more likely to be labeled if its neighbours are also labeled, and vice versa" [P1, first paragraph]. This depends completely on the labelling procedure, eg for Human Activity Recognition: when at random times people are asked what they do now, the labels dont follow this assumption. Also in the next paragraph: it is described that HAR data might have sequential bias, but it is not shown, what the properties of this bias are. Given that this is so important in the current manuscript, I'd like to see more insights:
1. What kind of bias is observed? I'd expect that people like to label in the beginning, and then forget about it later on. So that early in the sequence more labels are provided than later.
2. How can this biased be modelled? Please show some qualitative and quantitative results from different type of annotation/labelling procedures. Does the bias differ from domain to domain (eg medical vs human activity recognition)?
3. Does labelling bias depend on the annotator/person how labels the sequence?
4. Can the (obtained) propensity score be used for predicting labelling bias. How good is it at this? Or would a temporal smoothing over labels do equally well?

**Problem Setting**
- The problem setting assumes there are sets of sequences. Why is that? What defines such a single sequence?
- Can the setting be generalised to multi-class  classification? In that case a positive label for one class is a (strong indicator for) a negative label for all other classes.
- x_t seems to be a scalar-value, why is that? Can the model be used with more complex input (eg images)?

**Methodology**
- Figure 1 seems to contain an error: both the SPM and CN always take the learned representation of the feature sequence, not the feature sequence itself. It is also a rather complicated figure, if the only thing to show is that the networks are trained alternately.
- For my understanding it would be nice if the relation with related work is made here more clearly. Where does this method differ from the PN network? Only in the loss? If yes, which loss is used then?

**Experiments**
- I miss an ablation study for the effect of Rpu, PMC and OMC in the final objective (below Eq 3). What is the influence of adding PMC to the training pipeline? Or is Rpu alone sufficient?
- I miss experiments on multi-class datasets
- I miss the details on the datasets (type of features, number of examples, influence of subsampling etc).

**Conclusion**
To restate again, I think the observation of sequential bias is valid and it has to be present in many (sequential) labelling tasks. While in this paper some form of a solution is provided, I do miss the insights into this bias. I hope these can be provided during the rebuttal phase.

**Post-Rebuttal**
The authors have addressed some of my concerns during the rebuttal phase. Thanks! For me the major drawback of the current manuscript remains that 'sequential bias' is not really defined, and not really showed to exist in the datasets at hand. Only indirectly, by our new method performs better than existing methods. I think this should be improved, to increase the understanding of this topic.

---

> ### Author Response · Authors · 2020-11-21
> **Thank you for your insightful feedback**
>
> Thank you for giving us this detailed feedback on our work. We address your concerns below:
>
> Q: The assumption that an instance is more likely to be labeled if its neighbors are labeled is not reasonable in all use cases. In particular, in HAR participants are asked at random what their current activities are
>
> A:  First, thank you for alerting us that the quoted sentence from our paper can be misunderstood.  We will adjust the words to clarify.  It is not our intent to state that the only form that sequential bias can take is the simple case where if neighbors are labeled then a point is likely labeled; that would be merely one straightforward example of sequential bias. Instead, the reason we use a flexible recurrent network to model the sequential probability score is because the propensity function might be a complex function of the previous labels.
>
> Second, asking participants what activity they are performing at random times is NOT the standard method for in-the-wild HAR data collection studies. Rather, participants can label their activities at any time of their choosing.
>
> ---------------
>
> Q : Does the bias differ from domain to domain (eg medical vs human activity recognition)?
>
> A: In our paper we show results from two real-world in-the-wild data collections of both HAR data (ExtraSensory; collected in 2016 by San Diego) and medical data (RealityCommons, collected in 2008 by MIT). Clearly, our results confirm that our method improves over the alternate approaches which do not account for this sequential bias. This indicates that our approach’s  handling of the sequential bias arising in either of these real-world applications appropriately.
>
> --------------
>
> Q: Show qualitative and quantitative results from different type of annotation/labelling procedures.
>
> A: We have added in new section to our appendix that shows a quantitative analysis of the labeling in the Extra Sensory HAR study. Namely, we show a histogram of lengths of consecutive labeling in the Extra Sensory dataset. Importantly, this shows that the data does indeed follow our assumption that there are periods of time where the participants label consecutively and periods of time where they do not provide any labels, rather than the labeling being i.i.d. events.
>
> ---------------
>
> Q: The problem setting assumes there are sets of sequences. Why is that? What defines such a single sequence?
>
> A:  Correct,  such labeling data sets in many real-world situations including all the data sets we worked with consist of multiple sequences, not one long single sequence. For instance, in human activity recognition, when observing multiple people there is a separate sequence corresponding to the smartphone sensor data per each individual.  Thus for n participants we have n sequences put together into one data set.
>
> ---------------
>
> Q: Why do you not consider the multi-class setting and show results on this?
>
> A: We do not address the multi-label case as it does not fit our positive unlabeled problem setting. Namely, if we had reliable negative labels for one class (obtained from positive labels of another mutually-exclusive class), then the problem would be a semi-supervised learning task and not a positive-unlabeled task.
>
> However, we do apply this positive-unlabeled technique in the related multi-label setting (where there are multiple classes but they are not mutually exclusive), rather than the multi-class one. We can use positive unlabeled methods in this setting by performing binary decomposition. Indeed, all of the datasets in the experimental section other than Reality Commons are multi-label datasets that we apply binary decomposition on.
>
> ----------
>
> Q : How does the PN network differ from DeepSPU? Only in the loss function?
>
> A: Unlike our DeepSPU, the PN network does not have the Sequential Propensity Model component and it does not use DeepSPU’s loss function. Instead, PN network uses the logistic loss. Beyond these innovations, the base structure of components across these networks is the same.
>
> ----------
>
> Q: Why is there no ablation study for the effect of Rpu, PMC, and OMC?
>
> A: We in fact have done this exact ablation study, which was described in Section 5.4 and its results were shown in Figure 2 a of the submitted paper.
>
> ----------
>
> Q: Figure 1 seems to contain an error: both the SPM and CN always take the learned representation of the feature sequence, not the feature sequence itself
>
> R: This is not an error. Rather this is by design, namely, both models do take in a shared latent representation and not the raw input features of the data
>
> ---------
>
> Q: Is x_t a scaler? Can the method be applied to more complex data such as images?
>
> A: x_t is not a scalar. It can be a multivariate vector of arbitrary dimensionality, and could thus represent an image. In all of our experiments x_t is a high dimensional multivariate variable, usually around 50 dimensions for most datasets.

---

> > ### Comment · AnonReviewer6 · 2020-11-24
> > **Thanks for the clarification, still a few questions remain.**
> >
> > ## Sequential Bias
> > What I miss in the current manuscript is more understanding of the assumed sequential bias. Since it is so fundamental to this paper, I had expected / hoped for insights / experiments / analysis which directly show the presence of the sequential bias.
> > For example by fitting a particular distribution over the labels of the trainset (eg a Gamma distribution to model the time between two labels).
> > The current manuscript (and answers above), only show the bias **indirectly** via performance improvements on a classification task when using  a method which models some form of sequential bias.
> >
> > ## Remaining unclarity
> > >Q: Figure 1 seems to contain an error: both the SPM and CN always take the learned representation of the feature sequence, not the feature sequence itself.
> > >R: This is not an error. Rather this is by design, namely, both models do take in a shared latent representation and not the raw input features of the data
> >
> > I think I understand the flow of the network, however if both models take the shared latent representation, then it is unclear why why Figure 1 does include dotted arrows directly from the  `  feature sequence`  to the instantiated networks, by passing the Representation Network.

---

> > > ### Author Response · Authors · 2020-11-25
> > > **We added a new experiment for the propensity score and corrected Figure 1**
> > >
> > > Thank you for your continued feedback and time. We have added the new experiment you have suggested into our paper, in particular, into Section A.9 (appendix). It  demonstrates that our estimated propensity scores indeed matches the true propensity scores.
> > >
> > > To do this, we have created a new example where the true propensity score is known. We created subsequences of positive instances and subsequences of negative instances. The feature values for the positive instances and the negative instances were drawn from separate normal distributions. The positive subsequences were labeled using a gamma distribution to decide which positive instances receive labels, as you suggested.  We then trained DeepSPU on this data and extracted its learned propensity scores. A more detailed description of this new experiment has also been added to the revised paper.
> > >
> > > Further, as the new Figure 5 in the paper shows, the estimated propensity scores match the true propensity scores incurred by the gamma distribution almost perfectly. This confirms now empirically that we indeed learn the true propensity scores with our method.
> > >
> > > We apologize for the confusion regarding Figure 1. You are indeed correct that the arrows should go from the learned representation instead of the raw input sequence. We have revised the figure and include the revised figure in the paper.

---

### Official Review · AnonReviewer3 · 2020-11-07
**The paper presents a novel algorithm for binary classification of partially labeled inputs that come in a form of a sequence under a biased labeling mechanism. A novel loss function is derived by incorporating the propensity score in the standard PU loss formulation which is shown to reduce to the PN loss in expectation when the estimated q_i is equal to the true q_i.**

**Rating:** 5
**Confidence:** 5

**Review:**

Overall comments: Novel formulation and good experimental results. The method is claimed to be a general solution to PU settings with biased positives. However, there might be potential issues related to identifiability of the posterior from the labels. Mathematical writing lacks rigor.

Strengths:
1) The formulation is Novel.
2) Experiments show superior performance then the state of the art.

Weaknesses:
1) The algorithms for class prior estimation in PU learning are only derived for the no bias case. Thus class prior estimated using biased positives can be wildly incorrect. And consequently, it can introduce significant bias in the learnt posterior.
2) The new loss function is a function of the posterior and the propensity function which are both learnt from the data. As implied by the authors it can’t alone be used to learn the true posterior as there is a trivial solution to the loss which assigns q_i=1 for all i and g(x_1) = l_i. The authors add terms corresponding to the prior matching cost and observation matching cost to solve the issue. However, it seems that the observation matching cost is not helpful in that regard since it is minimum when q_i*g(x_i)=l_i and consequently q_i = 1 and g(x_i) = l_i.
3) In general, it seems that q_i and posterior cannot be uniquely identified from the labeled data. The true posterior is the minimizer of the new PU loss only if the estimated q_i indeed corresponds to the true labeling mechanism. Is it not possible that the both estimated q_i and the estimated posterior are far away from the true labeling mechanism and the true posterior, respectively? In my opinion, a consistent estimate of the true posterior is not feasible, unless assumptions are made on the nature of the bias.
4) Lack of rigor in mathematical writing: g_\theta(X)=Py(Y|X) implies that  g_\theta is a function of the sequence, but used as the posterior at a given x in the sequence. It would be better to have separate notations for the theoretical quantities and their estimates. In equation 2 shouldn’t it be Bern(Pr(\hat{y}=1|x)). G_\theta is defined twice, as the posterior probability and also as the output of the model. In reality it is the output of the model and the probability is what it is intended to estimate. OMC is not just a function of X it is also a function of the label. Please define OMC completely as a summation over the dataset. J is used as a function of g_\theta and also as a function of Q_\phi.  q_i is used in two different ways: one where y_i is given and the other where y_i is not given.
5) It will be useful to mention that the loss functions belong to the proper loss family so that g(x) indeed corresponds to the posterior [1].
6) Please give details on how are hyperparameters \lambda_1 and \lambda_2 are tuned.

[1] Reid MD, Williamson RC. Composite binary losses. The Journal of Machine Learning Research. 2010 Dec 1;11:2387-422.

---

> ### Author Response · Authors · 2020-11-20
> **Thank your for your insightful review**
>
> Thank you for spending the time to provide feedback on our work. We address your concerns below:
>
> -------------------
>
> Q: DeepSPU relies on knowing the class prior, but all class prior estimation methods require unbiased data
>
> A: This is incorrect. Jain et al. [1] have recently proposed an approach for estimating the class prior from biased positive and unlabeled data, and this will be deployed for use with our method in cases where the class prior is unknown.
>
> -------------------
>
> Q: The observation matching cost is not helpful because the trivial and incorrect solution of g(x_i) = l_i and q_i = 1 would minimize it
>
> A: This is the precise motivation for the prior matching cost, which directly mitigates this problem. Due to the prior matching cost, if g(x_i) only predicts l_i, then it would then predict too few positive instances and incur a high cost from the prior matching cost. As g(x_i) can not predict only only l_i without incurring high cost, consequently the optimal solution will NOT be for q_i to just predict 1.
>
> -------------------
>
> Q: As the estimate of q_i depends on the posterior and vice versa, can’t both the estimated posterior and the estimated q_i converge to a far-off incorrect solution?
>
> A: This is why we begin training the posterior estimator using the SCAR assumption: So that we can get reasonably good values for the posterior that will in turn lead to reasonably good estimates of q_i, which are then iteratively improved in turn
>
> -------------------
>
> Q: The paper paper could benefit from clearer notation in parts.
>
> A: Thank you for helping us improve the writing. We will take your suggestions and modify the document.
>
> -------------------
>
> Q: What loss function does g(x) use?
>
> A:  We use the sigmoid loss, as suggested by Kiryo et al. [2].
>
> -------------------
>
> Q: What values are used for hyperparameters \lambda_1 and \lambda_2?
>
> A: These hyperparameters are set to 1 in all experiments. We will state this explicitly in the paper.
>
>
> -------------------
>
> [1] Jain, Shantanu, et al. "Class Prior Estimation with Biased Positives and Unlabeled Examples." AAAI. 2020.
> [2] Kiryo, Ryuichi, et al. "Positive-unlabeled learning with non-negative risk estimator." Advances in neural information processing systems. 2017.

---

> > ### Comment · AnonReviewer3 · 2020-11-25
> > **Thank you for the response to my questions. Here are my follow up quetions**
> >
> > Yes its true that  Jain et al. [1] contains a algorithm to estimate the class prior in the bias setting. However, they address bias under three assumptions. Since you are using that class prior, wouldn't it imply that you are implicitly making those assumptions. Was the class prior used in you experiments estimated using  Jain et al. [1]?
> >
> > In the paper it is implied that the observation matching cost also helps mitigate the bias. Is that incorrect? If yes, please make corrections in the paper.
> >
> > I agree that the class prior matching cost would promote g(x_i) to predict more positives, but how does that solve arbitrary bias? Solving arbitrary bias would mean that the distribution of positives implied by the estimated posterior matches the true distribution of positives. Now, there can be two different posterior estimates that lead to the same class prior but imply different distribution of positives. In my opinion matching the class prior is not sufficient to get to the correct posterior. A formal proof would be required to show that arbitrary bias in positives can be rectified. If this cannot be shown as a proof, the authors should not claim that arbitrary bias can be corrected by their approach. It would be more accurate to say that the bias it mitigated to some extent. Further, it would be helpful to add a disclaimer that the bias might not be completely mitigated.
> >
> > Why would starting at SCAR posterior guarantee that the algorithm would converge to the true posterior?

---

> > > ### Author Response · Authors · 2020-11-25
> > > **Thank you for your continued feedback**
> > >
> > > It is not our intent to claim that either the OMC or PMC solve for arbitrary bias. Instead, they are intended to stop the model from collapsing into certain incorrect solutions. The PMC stops the classification network from simply predicting the likelihood of labeling rather than the true posterior distribution, but the PMC cost alone is not enough to train the classifier because it can be minimized so long as any X% of the instances are predicted as positives, where X% is the class prior. The OMC makes it so that this X% can not be completely random, as the observed label positives should be among the X% predicted as positives. We refer to the OMC and PMC as regularization terms because we do not intend for them alone to solve the problem. They are regularization terms because a correct solution should incur no costs from these terms, but minimizing these costs is not enough to imply the solution is correct.
> > >
> > > This is a similar issue to the one faced by Bekker et al.'s SAR EM[1] approach that we compare against, which attempts to address feature-level bias. Their EM approach likewise can not guarantee that the learned posterior exactly matches the true posterior, as the trivial solution of the classifier just predicting the observed labels and the propensity score always predicting 1 would maximize the probability of the observed data. They attempt to avoid this trivial solution by dampening the propensity score so it can not always predict 1 (and thus the classifier cannot simply predict the labels), but the value to dampen the propensity scores by is arbitrary. Our intent with the PMC and OMC is to avoid this same trivial solution, but instead of picking an arbitrary value to dampen the propensity scores by we instead encourage our solution to have properties that we know the correct solution should have.
> > >
> > > Likewise, we do not intend to claim that staring with the SCAR assumption to pretrain the classifier will guarantee that we will converge to the true posterior in every situation, nor do we claim to have a proof for this. What we do find is that starting with this "good" but incorrect initial estimate of the propensity score empirically leads to a better solution than when we apply the leading approaches to data with a sequential bias. Additionally, the new experiment added in Section A.9 of the appendix empirically shows that our estimated propensity scores are very close to the true propensity scores.
> > >
> > > We are thus happy to reword our claims so that it is clear that we are mitigating the effects of sequential bias, but do not prove that our solution is optimal in all cases.
> > >
> > > [1] Jessa Bekker et al., "Learning from positive and unlabeled data under the selected at random assumption", Journal of Machine Learning Research 2018.

---

### Decision · Program_Chairs · 2021-01-07
**Final Decision**

**Decision:**

Reject

**Comment:**

Dear Authors,

Thanks for your detailed feedback to and even communications with the reviewers. Your additional input certainly clarified some of the concerns raised by the reviewers and also improved their understanding of your work.

However, we still think that the notion of sequential bias is unclear, and the authors overclaim what they have done.
For these reasons, this paper cannot be recommended for acceptance.
I hope that the detailed feedback from the reviewers will help improve this work for future publication.